# Diversity of the Bacterial Community Associated with Hindgut, Malpighian Tubules, and Foam of Nymphs of Two Spittlebug Species (Hemiptera: Aphrophoridae)

**DOI:** 10.3390/microorganisms11020466

**Published:** 2023-02-13

**Authors:** Anita Nencioni, Roberta Pastorelli, Gaia Bigiotti, Maria Alexandra Cucu, Patrizia Sacchetti

**Affiliations:** 1Department of Agricultural, Food, Environmental and Forestry Science and Technology (DAGRI), University of Florence, Piazzale delle Cascine 28, 50144 Florence, Italy; 2Research Center for Agriculture and Environment, Consiglio per la Ricerca in Agricoltura e l’Analisi dell’Economia Agraria (CREA-AA), Via di Lanciola 12/A, 50125 Florence, Italy

**Keywords:** facultative symbionts, juvenile stages, *Lepyronia coleoptrata*, *Philaenus spumarius*, *Xylella fastidiosa* vectors

## Abstract

Spittlebugs are xylem-sap feeding insects that can exploit a nutrient-poor diet, thanks to mutualistic endosymbionts residing in various organs of their body. Although obligate symbioses in some spittlebug species have been quite well studied, little is known about their facultative endosymbionts, especially those inhabiting the gut. Recently, the role played by spittlebugs as vectors of the phytopathogenetic bacterium *Xylella fastidiosa* aroused attention to this insect group, boosting investigations aimed at developing effective yet sustainable control strategies. Since spittlebug nymphs are currently the main target of applied control, the composition of gut bacterial community of the juveniles of *Philaenus spumarius* and *Lepyronia coleoptrata* was investigated using molecular techniques. Moreover, bacteria associated with their froth, sampled from different host plants, were studied. Results revealed that *Sodalis* and *Rickettsia* bacteria are the predominant taxa in the gut of *P. spumarius* and *L. coleoptrata* nymphs, respectively, while *Rhodococcus* was found in both species. Our investigations also highlighted the presence of recurring bacteria in the froth. Furthermore, the foam hosted several bacterial species depending on the host plant, the insect species, or on soil contaminant. Overall, first findings showed that nymphs harbor a large and diverse bacterial community in their gut and froth, providing new accounts to the knowledge on facultative symbionts of spittlebugs.

## 1. Introduction

Endosymbioses are widespread in insects and affect many aspects of their biology, ecology, and evolution [1,2]. Among heritable intracellular bacteria, three main categories of endosymbionts can be recognized: Primary symbionts (P-symbionts), secondary symbionts (S-symbionts), and reproductive manipulators [3].

P-symbionts are obligate mutualists inhabiting specialized cells called bacteriocytes, which may form organs known as bacteriomes [2,3]. They are vertically transmitted and provide essential nutrients to the host [4]. Since the association between P-symbionts and their hosts originates from an ancient infection, usually all the descendants of the infected ancestor have coevolved with the same P-symbiont [4].

On the other hand, S-symbionts and reproductive manipulators are facultative endosymbionts that can colonize several organs or even reside extracellularly in the hemocoel [3,4]. S-symbionts usually enhance the host fitness, for example, offering protection against stress or natural enemies, while reproductive manipulators could be considered as parasites, since they affect host reproduction to favor its own spread [3].

Sap-feeding insects, as those belonging to the sub-order Auchenorrhyncha, are known to host large communities of symbionts and bacteriome-associated mutualistic organisms, implicated mainly in the provision of nutrient lacking in the diet [4,5]. Particularly, the Bacteroidetes *Candidatus* Sulcia muelleri Moran et al. 2005 (Flavobacteriia: Blattabacteriaceae) has long been recognized as the P-symbiont of leafhoppers, spittlebugs, cicadas, and other Auchenorrhyncha species [6]. In many cases, *Ca*. Sulcia muelleri is coupled with another S-obligate symbiont that occupies a specific area of bacteriomes. Typically, co-resident endosymbionts show a complementary role in the biosynthesis of essential amino acids [7,8].

The Cercopoidea superfamily comprises xylem-feeding insects that are commonly known as froghoppers or spittlebugs, due to the habit of nymphs to develop inside a self-produced foam nest. This foam is formed by the excretion of the alimentary canal, mainly composed by metabolized xylem sap, added with mucopolysaccharides and proteins produced by specialized cells of the Malpighian tubules [9,10,11]. Nymphs introduce air bubbles in this mixture through telescopic movements of the abdomen, giving the typical frothy state to their excreta. The foam protects nymphs against dehydration, temperature fluctuations, and natural enemies [12,13,14].

Spittlebugs are spread in many terrestrial ecosystems and some species are known to be important phytophagous [12,15,16,17]. The meadow spittlebug *Philaenus spumarius* L. 1758, is currently considered as a major pest in Europe due to its competence to transmit the xylem-inhabiting harmful bacterium *Xylella fastidiosa* (Wells et al. 1987) subsp. *pauca* ST53 (Gammaproteobacteria: Xanthomonadaceae) [12,18], which is the causal agent of the Olive Quick Decline Syndrome [19].

In spittlebugs, *Candidatus* Zinderia insecticola and a *Sodalis*-like bacterium, allied to *Sodalis glossinidius* (Dale and Maudlin 1999) (Gammaproteobacteria: Pectobacteriaceae), are the two bacteriome-associated endosymbionts known, to date, to be co-residents of *Ca*. Sulcia muelleri [5,8]. In particular, *Ca*. Zinderia insecticola is associated with most of the spittlebugs, while the *Sodalis*-like bacterium is an endosymbiont of the species belonging to the tribe Philaenini, such as *P. spumarius* [5].

In contrast to P-symbionts, the microbial community of spittlebugs’ facultative endosymbionts has received less attention. The genera *Rickettsia*, *Arsenophonus*, *Hamiltonella,* and *Wolbachia* are reported to be facultative symbionts of *P. spumarius* and other Philaenini species [20,21]. However, studies on the anatomical localization of these bacteria are lacking. Similarly, functions of mutualistic microorganisms other than P-symbionts are still little explored in spittlebugs. For instance, the hypothesis that the gut microbiota of aphrophorid nymphs could play a role not only in the hydrolysis and assimilation of food, but also in the foam production, has never been tested [22]. Furthermore, information on the microbial community associated with the froth is scarce, and the origin and function of foam-inhabiting microorganisms are still poorly known. Recently, most of the bacteria found in the froth of *Mahanarva fimbriolata* (Stål 1854) (Hemiptera: Cercopidae) have been identified as alpha-Proteobacteria, microorganisms which presumably might play a defensive role against different natural enemies, although evidences supporting this hypothesis are lacking [23].

Since it has been recognized as the potential of novel, effective control strategies that could be developed on the basis of the knowledge on pests’ microbiome [24], the study of *P. spumarius* facultative endosymbionts appear to be of great importance, in order to manage the *X. fastidiosa* emergency in Europe. Currently, spittlebug juveniles are considered as the main target of applied control strategies due to their reduced mobility [25]. Hence, analyzing the structure of microbial community associated with nymphs’ gut and froth could be useful to develop effective and sustainable control strategies against aphrophorid nymphs.

As a result, in this study, the endosymbionts harbored in the mid- and hindgut of *P. spumarius* nymphs were explored using molecular procedures. Moreover, nymphs of the spittlebug *Lepyronia coleoptrata* (Linnaeus 1758) (Hemiptera: Aphrophoridae) were examined to explore and compare the gut microbiota of a non-Philaenini species. Finally, in both species, the structure of the bacterial community associated with the foam and the Malpighian tubules involved in the froth production were studied.

## 2. Materials and Methods

### 2.1. Collection of Foam and Insect Samples

Foams and insect samples have been collected and processed following the procedures described for *M. fimbriolata*, after minor changes [23].

Fifth-instar nymphs of *P. spumarius* and *L. coleoptrata* were collected in the field from *Ranunculus* sp. and *Trifolium repens* L. (Fabaceae), respectively, and brought alive to the laboratory to dissect the alimentary canal (Table 1). Late-instar nymphs have been chosen since they produce a greater amount of foam and to facilitate insect dissections. Specimens were cold-anesthetized, externally sterilized in a 2% sodium hypochlorite solution, rinsed two times in sterile deionized water, and then dissected in a sterile 0.8% NaCl solution. Dissections were carried out using sterile tools, under a stereoscopic microscope in a laminar-flow hood.

For *P. spumarius*, the filter chamber linked to the conical segment, the posterior tubular midgut, and the Malpighian tubules were obtained. The ileum was added to the above-mentioned sections for *L. coleoptrata*, since the presence of bacteria in this part of the hindgut had been formerly evidenced [26]. Anatomical regions were located and cut apart according to their previously reported morphological descriptions [26,27]. Insect samples were formed by pooling five dissected portions of the same type; samples were stored in sterile 1.5 mL vials and maintained at −80 °C until processed.

Foam samples were collected directly in the field (Table 1) by means of sterile spatulas and microscope slides, using 95% ethanol and wearing gloves to prevent unintentional contamination. Each sample was formed by the foam produced by five fifth-instar nymphs, which were feeding on the same plant species. Three replicates per host plant species were collected in variable quantities from 2.5 up to 10 mL, depending on the spittlebug species and host plant. Each foam sample was maintained and treated separately, stored in sterile 15 mL vials, and maintained at −80 °C until processed.

### 2.2. Microbiological Analyses

Bacterial DNA extraction was carried out using the FastDNA^TM^ Spin Kit for Soil (MP Biomedicals, Santa Ana, CA, USA) according to the manufacturer’s instructions. Frozen insect samples were crushed and mashed with a sterile pestle prior to the beginning of the extraction procedure. Foam samples were thawed, cleaned from impurities by centrifugation (14,000× *g* for 5 min), and then treated with the extraction kit. To assess the extraction quality and integrity, DNA preparations were visualized by electrophoresis in 1% (*w/v*) agarose gel at 4 V/cm for 1 h in TAE buffer (89 mM Tris base, 89 mM boric acid, 2 mM EDTA; pH 8.3), stained with Xpert green DNA stain (GriSP Research Solutions, Porto, Portugal), and observed under UV light.

#### 2.2.1. Real-Time PCR

Quantitative real-time PCR was performed to quantify the bacterial biomass in each gut sample. Amplification reactions were carried out in an ABI StepOne^TM^ Real-Time PCR system (Applied Biosystems, Waltham, MA, USA) in a 10 μL mixture containing 1X BlasTaq^TM^ qPCR MasterMix (Applied Biological Materials Inc., Richmond, BC, Canada), 400 nM of each primer (341F and 515R [28]), and 1 μL of template DNA. The amplification conditions consisted of incubation at 95 °C for 10 min, followed by 40 cycles of denaturation at 95 °C for 15 s, annealing at 55 °C for 60 s, and extension at 72 °C for 45 s. All samples were carried out in triplicate in optical 96-well plates along with the negative control and standard curve. Standard curve was created for absolute quantification of 16S rRNA gene using a plasmid containing the target gene fragment from *Pseudomonas* sp. DSM1650 10-fold diluted from 3.60 × 10^7^ to 3.60 × 10^3^ gene copy numbers μL^−1^. Fluorescent light outputs were collected during each elongation step and analyzed with the ABI StepOne Real-Time PCR system SDS software v 2.3 (Applied Biosystems). Melting curves were generated after amplification by increasing the temperature of 0.5 °C every 30 s from 65 to 95 °C, in order to verify the absence of primer dimers or artifacts.

#### 2.2.2. Denaturing Gradient Gel Electrophoresis

Denaturing gradient gel electrophoresis (DGGE) analysis was performed using the universal primers 986F-GC and 1401F [29], designed to amplify the V6-V8 region of the 16S rRNA bacterial gene, in order to explore the composition of the bacterial community associated with the gut, the Malpighian tubules, and the foam of the two spittlebug species. Amplification reactions were carried out in a T100 Thermal Cycler (Bio-Rad Laboratories, Watford, UK) in a 25 μL volume containing 1X Xpert Taq Reaction Buffer (GRISP Research Solution), 1.5 mM MgCl_2_, 250 μM of deoxynucleotide triphosphate (dNTPs), 400 nM of each primer, and 1 U of Xpert Taq DNA (GRISP Research Solution). The reaction conditions consisted of an initial denaturation of 94 °C for 5 min followed by 35 cycles of 94 °C for 30 s, annealing at 55 °C for 30 s, extension at 72 °C for 45 s, and a final extension of 72 °C for 10 min. Successively, the amplification products were loaded onto a 6% polyacrylamide gel (acrylamide/bis 37.5:1), with a 42–68% linear denaturing gradient increase in the electrophoretic run direction, and obtained with a 100% denaturing solution containing 40% formamide (VWR, Radnor, PA, USA) and 7 M Urea (Promega, Madison, WI, USA). The gels were carried out in a D-Code System (Bio-Rad) for 18 h in 1X TAE buffer at constant voltage (80 V) and temperature (60 °C), and stained with SYBR^®^GOLD (Molecular Probes, Eugene, OR, USA) diluted 1:1000 in 1X TAE. DGGE images were digitally captured under UV light using a Chemidoc XRS apparatus (Bio-Rad).

#### 2.2.3. Sequence Analysis

The dominant DGGE bands were aseptically excised from gels and sequenced to taxonomically identify bacterial symbionts (Appendix A). The middle portion of each band was placed in 25 μL of distilled water and stored at −20 °C overnight. Successively, DNA fragments were eluted from the gel through freezing and thawing and 1 μL of each elution was used as a template in an amplification reaction carried out as previously described for DGGE analysis. The re-amplified PCR products were purified using PureLinkTM Quick PCR Purification kit (Invitrogen-Life Technologies, Carlsbad, CA, USA) and sent to the center CIBIACI (Centro Interdipartimentale di Servizi per le Biotecnologie di Interesse Agrario, Chimico, Industriale) of the University of Florence for the sequencing service. The obtained sequences were edited using Chromas Lite software (v 2.1.1; Technelysium Pty Ltd., South Brisbane, Australia; http://www.technelysium.com.au/chromas_lite.htm, accessed on 21 July 2022) to verify the absence of ambiguous peaks and convert them to the FASTA format. The DECIPHER’s Find Chimeras web tool (http://decipher.cee.wisc.edu, accessed on 21 July 2022) was used to uncover chimeras hidden in the 16S rDNA sequences. Nucleotide sequences were compared against all sequences stored within the NCBI database using the Web-based BLAST tool (http://www.ncbi.nlm.nih.gov/BLAST, accessed on 21 July 2022) to find closely related nucleotide sequences. Taxonomical identification was achieved by means of different sequence similarity thresholds as described by Webster et al. [30]. The nucleotide sequences were deposited in the GenBank database under accession numbers OP012727-OP012755.

### 2.3. Statistical Analysis

One-way analysis of variance (ANOVA) followed by Fisher least-significant difference (LSD) were applied to analyze data obtained from the real-time PCR using Statistica 12.0 software (StatSoft, Palo Alto, CA, USA).

The Gel Compare II software v 4.6 (Applied Maths, Sint-Martens-Latem, Belgium) was used to analyze the foam DGGE and to convert it into a matching table based on the presence/absence and intensity of bands within each banding pattern, in order to be imported into the freely available PAST 4.03 software for subsequent multivariate statistical analysis [31]. Non-metric multidimensional scaling (nMDS) analysis and one-way analysis of similarity (ANOSIM) were performed using the Bray-Curtis distance measure and 9999 permutational tests, in order to visualize the similarity/dissimilarity of bacterial communities hosted in the collected foams in a two-dimensional space and to determine the extent of similarity/dissimilarity according to the different plant species, respectively. The accuracy of the nMDS plot was determined by calculating a 2D stress value. An ANOSIM R value of 1 indicates that the bacterial communities of foam collected from each plant species are more similar to each other than to any sample from another plant species, whereas an R value of 0 indicates that there is as much variation within a group as among the groups being compared. More specifically, 0.5 < R values < 0.75 were interpreted as separated but overlapping [32].

## 3. Results

### 3.1. Bacterial Biomass Quantification

A total of 20 *P. spumarius* and 15 *L. coleoptrata* nymphs and a quantity of about 40 and 30 mL of foam, respectively, were analyzed. Bacterial DNA was extracted from all the samples. The richness in endosymbionts (quantified by the real-time PCR) was expressed as the number of 16S rRNA gene copies (Figure 1). The larger bacterial biomass (6.00 × 10^5^ ± 6.86 × 10^5^ 16S rRNA gene copies) was harbored by the filter chamber joined to the conical segment of *L. coleoptrata*. Compared to *P. spumarius*, *L. coleoptrata* showed a higher average content of bacteria in the posterior tubular midgut (1.08 × 10^5^ ± 1.52 × 10^5^ 16S rRNA gene copies) and in the Malpighian tubules (1.38 × 10^5^ ± 2.17 × 10^5^ 16S rRNA gene copies). Even the ileum, which was the poorest part in terms of symbionts abundance (2.70 × 10^4^ ± 2.44 × 10^4^ 16S rRNA gene copies), contained a mean bacterial load higher than measured for the posterior tubular midgut of *P. spumarius* (2.40 × 10^4^ ± 1.80 × 10^4^ 16S rRNA gene copies). The greatest abundance of bacteria in *P. spumarius* was in the Malpighian tubules (7.28 × 10^4^ ± 5.57 × 10^4^ 16S rRNA gene copies).

Statistical analyses showed that the filter chamber linked to the conical segment of *L. coleoptrata* was the only studied portion that contained a number of symbionts significantly higher than the other considered anatomical parts, including those of *P. spumarius*. No other significant differences have been highlighted among the analyzed gut portions and organs within each species (Figure 1).

### 3.2. Gut Bacterial Community Composition

The DGGE profiles of gut dissections and Malpighian tubules of both *P. spumarius* and *L. coleoptrata* revealed three dominant bands, with two of them shared by the two spittlebugs, suggesting the presence of common endosymbiont species (Appendix A). Other less prominent bands were revealed for both insects. Sequencing analysis of excised DGGE bands of *P. spumarius* gut samples and Malpighian tubules revealed the sharing of sequence identity with a *Sodalis*-like bacterium allied to *S. glossinidius* (98% similarity to GenBank accession number LN854557)*,* a species belonging to the Enterobacteriaceae family (*Escherichia coli*: 100% similarity to GenBank accession number MN083301), and a *Rhodococcus* species (*Rhodococcus gingshengii:* 100% similarity to GenBank accession number MN826591). The gene sequencing disclosed the presence of a *Rickettsia*-endosymbiont (*Rickettsia bellii*: 98.8% similarity to GenBank accession number KU586119), the Enterobacteriaceae species (*Escherichia coli*: 100% similarity to GenBank accession number MN083301), and a *Rhodococcus* bacterium *(Rhodococcus gingshengii:* 100% similarity to GenBank accession number MN826591) as likely putative endosymbionts of *L. coleoptrata* (Appendix A).

### 3.3. Foam Bacterial Community Composition

DGGE profiles of foam samples showed the presence of several dominant bands, along with other fainter ones (Appendix A). Some conspicuous bands were common to both *P. spumarius* and *L. coleoptrata* foam samples. All bands selected for sequencing showed high similarity to the species belonging to Proteobacteria, with a majority of alpha-Proteobacteria, such as *Brevundimonas mediterranea* (99.7% similarity to GenBank accession number MK250497), *Devosia oryziradicis* (97.2% similarity to GenBank accession number CP068047), and several Rhizobiaceae species. Members of Rhizobiaceae detected in the froth, displayed a distinct discrepancy in the association between insect and bacteria species. Indeed, *P. spumarius* foam samples seemed to harbor bacteria species referring to the genus *Ciceribacter*, while spittles of *L. coleoptrata* contained mainly other genera (Table 2).

The nMDS, used to evaluate the DGGE profiles obtained through the analysis of the foam samples of the two spittlebug species, showed a certain diversity in the composition of the microbiome associated with the froth. As shown in Figure 2, bacterial communities could be grouped using the host plant as a parameter, indicating that foams collected from the same plant species harbored a similar microbial community. However, the structure of microbial assemblages associated with foams collected from different plant species appeared to be distinct, with the only exception of froth samples collected from *Vicia sativa* L. (Fabaceae) and *Euphorbia cyparissias*, that could be grouped together. Moreover, nMDS analysis displayed that the bacterial communities of the froths collected from *Vicia sativa* showed almost an identical structure, since they could be graphically overlapped. Finally, results obtained with ANOSIM (R = 0.7689; *p* = 0.0001) allowed for a clarification on the fact that the host plant and insect species significantly affected the bacterial community structural diversity. On the contrary, the effects of the sampling site seemed to be negligible, since foams collected in the same locality harbored different microbiomes, while froths collected from *V. sativa* and *E. ciperacea*, with similar microbial structure/composition, originated from two different areas.

## 4. Discussion

The posterior part of the insect’s alimentary canal is often characterized by the presence of microbial endosymbionts that are typically located in the hindgut [33]. The wide range of functions performed by these microorganisms—well summarized by Engel and Moran [34]—is one of the key elements that have allowed for the successful evolution and spread of insect species on earth.

Although the presence of bacterial cells in adult spittlebug’s gut has already been highlighted by microscopy studies [26,27], and gut endosymbionts have been reported for other Auchenorrhyncha species [35,36], knowledge on the gut microbiota of Aphrophoridae species is still inadequate.

Results from the present study showed that *P. spumarius* and *L. coleoptrata* nymphs harbor a large and diverse community of bacterial endosymbionts in their gut and in the Malpighian tubules, with some predominant species revealed by DGGE profiles.

A species closely related to *S. glossinidius* was recurrently found in the examined *P. spumarius* gut samples, remarking the importance of this symbiont for spittlebugs. Indeed, the *Sodalis*-like bacterium inhabiting the bacteriomes of Philaenini species, along with *Ca*. Sulcia muelleri, can provide hosts with essential amino acids and other important biomolecules, despite its reduced genome [8]. Particularly, the potential capability to synthetize glutamate from glutamine (one of the main sources of amino acids in xylem sap) is remarkable, since it is complementary with the probable capacity of *Ca*. Sulcia muelleri to synthetize 2-oxoglutarate, a molecule involved in the Krebs cycle, starting from glutamate [8]. Moreover, the detection of *Sodalis* endosymbiont outside of bacteriomes underlined the importance of its biosynthetic capabilities, marking the key role of this bacteria in the provision of essential nutrients lacking in the host diet.

Since the relationship between *Sodalis* and members of the tribe Philaenini appears to be an important aspect in the evolutionary success of these spittlebugs, further investigations are needed to clarify the precise localization of this endosymbiont and its functions in *P. spumarius.*

Our findings on *L. coleoptrata* gut microbiome, highlighted the presence of a *Rickettsia* bacterium (98.8% similarity to *Rickettsia bellii*) both in the gut and Malpighian tubules. The *Rickettsia* group comprises intracellular bacteria well known as vertebrate pathogens, although many species belonging to this genus live in non-pathogenic association with vertebrate and invertebrate animals [37]. In arthropods, members of the genus *Rickettsia* usually act as vertically-transmitted facultative symbionts that play a major role as reproductive manipulators, even if the variety of their effects on the hosts is still little documented [37,38,39,40]. Different *Rickettsia* species have been found in several hemipterans [41,42], including Auchenorrhyncha [20,43,44,45,46]. In these cases, *Rickettsia* spp. are involved in reproductive manipulation [42,47], increase in the host fitness [47], thermotolerance [48], and protection against pathogens [49]. Since we detected *Rickettsia* sp. harbored by digestive organs of *L. coleoptrata*, we can speculate that this symbiont might be involved in digestion processes, as proposed for some cicada species [45]. Otherwise, the symbiont could even be implied in the production of the froth, since a possible contribution of this bacterium was suggested in the production of the gelling saliva delivered by *Bemisia tabaci* (Gennadius 1889) (Hemiptera: Aleyrodidae) to allow for stylets penetration [50].

According to our research, Enterobacteriaceae species and a member of Actinobacteria of the genus *Rhodococcus* are putative endosymbionts of both *P. spumarius* and *L. coleoptrata* nymphs.

The association between insects and Enterobacteriaceae is demonstrated as wide-spread and diverse [33]. Phylogenetic studies have highlighted that several insect endosymbionts are closely related with this bacteria family, indicating the presence of specific traits that allow for Enterobacteriaceae to infect and establish inside of the insect hosts [51]. Beneficial effects originated by these mutualistic relationships vary from the increase in resistance to stress and parasitism to the extension of the host plant range [52]. Moreover, the family Enterobacteriaceae comprises diazotroph organisms that allow for nitrogen (N_2_)-fixation in insects [53]. Since spittlebugs feed on nutrient-poor and nitrogen-deficient diet, it is likely that the Enterobacteriaceae inhabiting their gut are involved in N_2_-fixation processes, providing the host with N-usable sources [52].

The Actinobacteria genus *Rhodococcus* includes several species, often found in the soil, that are able to degrade many toxic chemicals [54]. Even *R. gingshengii* was first described as a fungicide degrading-bacterium, when it was isolated from carbendazim-contaminated soils and its capability to reduce this noxious compound was assessed [55]. Members of the genus *Rhodococcus* were also reported as symbionts of several insects, as in the case of *Rhodococcus rhodnii*, a gut symbiont of the bug *Rhodnius prolixus* (Stål 1859) (Hemiptera: Reduviidae) that supplies its host with B vitamins [56,57]. However, this is only one of a few recognized nutritional-based associations between Actinobacteria and insects, since this group appears to be generally involved in defensive mechanisms, producing secondary metabolites with antibiotic effects [58].

Regarding Auchenorrhyncha, *Rhodococcus* was previously detected in two cicada species, *Platypleura kaempferi* (Fabricius 1794) (Hemiptera, Cicadidae) and *Meimuna mongolica* (Distant 1881) (Hemiptera, Cicadidae) [45], and in the leafhopper *Homalodisca vitripennis* (Germar 1821) (Hemiptera, Cicadellidae) [59]. Nevertheless, the nature of the association between *Rhodococcus* bacteria and their Auchenorrhyncha hosts is unknown. Therefore, the role played by *Rhodococcus* sp. in spittlebugs should be further investigated.

The bacteria species discovered in the gut of *P. spumarius* and *L. coleoptrata* were identified also in the Malpighian tubules of both species. In insects, Malpighian tubules are the main excretory organs, but in Cercopoidea species, they acquire a secretory function thanks to cytological modifications that allow for the production of some components of the froth, mainly mucopolysaccharides and proteins [60]. Through ultrastructural studies, bacterial cells were evidenced in Malpighian tubules of *Aphrophora obliqua* (Uhler 1896) (Hemiptera, Aphrophoridae) adults, showing morphological similarity to those present in the gut of the same species [61]. In the same research, young juveniles did not display these microorganisms, leading to the hypothesis of an inhibitory function of the secretions of Malpighian tubules in nymphs [61]. Our results corroborate the evidence that symbiotic bacteria can be associated with Malpighian tubules in Cercopoidea species. In contrast to what was observed in *A. obliqua*, we detected bacteria in Malpighian tubules of nymphs of both species, *P. spumarius* and *L. coleoptrata,* presumably since we dissected fifth-instar juveniles, which are quite close to the emergence and that can show dissimilar features from younger stages.

The complex composition of the froth produced by spittlebug nymphs offers a rich and diverse substrate for numerous microorganisms that could contribute to create a suitable environment for the insect and to produce direct beneficial effects for the nymph itself. To date, the microbial community of spittlebug froth has received scarce attention and information on this topic is almost absent. Our results suggest that the class of alpha-Proteobacteria is the predominant group of bacteria residing in the examined spittlebug froths, as already highlighted for the foam of the Neotropical member of Aphrophoridae *M. fimbriolata* [23]. Alpha-Proteobacteria form a heterogeneous group of microorganisms, inhabiting several terrestrial and aquatic ecosystems. Furthermore, they are involved in many associations with eukaryote organisms, including insects [62].

*Brevundimonas mediterranea* is an alpha-Proteobacterium recently isolated and described from samples of Mediterranean Sea water [63]. Members of the genus *Brevundimonas* are reported to be capable of degrading environmental contaminants [64,65,66] and mycotoxins [67], but also to enhance the growth of plants and microalgae when involved in symbiotic association [68,69,70].

In our froth samples, bacteria attributable to *Brevundimonas mediterranea* appear to occur independently from insect species, populations, and localities. We can speculate that this mutualistic association could be of great importance for spittlebug nymphs.

Members referring to the Rhizobiaceae family appeared to be abundant in the analyzed froth samples of *P. spumarius* and *L. coleoptrata*. The family Rhizobiaceae includes phenotypically diverse organisms, ranging from N_2_-fixing legume symbionts to plant pathogens, bacterial predators, and other soil bacteria. Their involvement in endosymbioses with insects has been documented for several species, including those belonging to the Hemiptera order, for instance, *Ca*. Hodgkinia cicadicola (McCutcheon et al. 2009) (*Alphaproteobacteria*) is reported to be the co-resident symbiont of *Ca*. Sulcia muelleri in some cicada species [71,72]. One of the recognized functions of Rhizobiales in insects is the fixation of gaseous N_2_ into more usable forms [53]. Our analyses displayed the presence of bacteria, such as *C. azotofigens* or *Rhizobium* sp. in all the froth samples, but the same species were not evidenced in gut samples of both spittlebugs. Therefore, the most likely infection source for the foam might be the soil or the host plant, since members of Rhizobiaceae typically inhabit these ecological niches. Due to their abundance, probably, the spittlebugs’ foam represents a suitable substrate for the survival and growth of the bacteria of the family Rhizobiaceae; however, if any possible interaction exists with froth producing insects, it remains unclear.

The structure of the remaining part of the bacterial community associated with the spittle mass depends on both the insect and host plant species, as shown by the nMDS analyses. The major source of bacteria associated with the froth might be the plant on which the nymph feeds. Differences in dietary habits of the two studied spittlebug species could explain the observed diversity.

Findings highlighted by this study suggest that the froth is a complex environment harboring both specific mutualistic bacteria, which might have defensive functions (e.g., *Brevundimonas*) and other arbitrary species which are contaminant, originating from the soil or host plant as it was already highlighted for the spittlebug *M. fimbriolata* [23] and for another animal secretion, i.e., the mucus of the garden snail *Cornu aspersum* (O.F. Muller 1774) (Mollusca, Helicidae) [73].

Although the results obtained from DGGE, and subsequently, the sequencing of dominant bands could be very useful for a preliminary overview of bacterial communities, the complete sequencing of extracted DNA via next-generation sequencing (NGS) analysis will be more informative, in order to provide a detailed description of spittlebug facultative endosymbionts and of the bacteria hosted by the froth. Moreover, ultrastructural studies would be necessary to assess the exact localization of bacteria in the gut and Malpighian tubules of spittlebug nymphs.

More focused research on symbionts of spittlebugs could clarify specific functions played by bacteria detected in the nymphs’ organs and unveil their roles in the relationship with their hosts. Moreover, additional investigations could disclose the bacterial involvement in the production of the froth.

In conclusion, our study has allowed for a preliminary exploration of the complex bacterial community associated with the gut, the Malpighian tubules, and the foam of the nymphs of two Aphrophoridae species. Moreover, the study provides the basis for further and more deepened investigations aimed at improving the knowledge on this topic and developing effective and more sustainable control strategies against spittlebug vectors of *X. fastidiosa*.

## Figures and Tables

**Figure 1 microorganisms-11-00466-f001:**
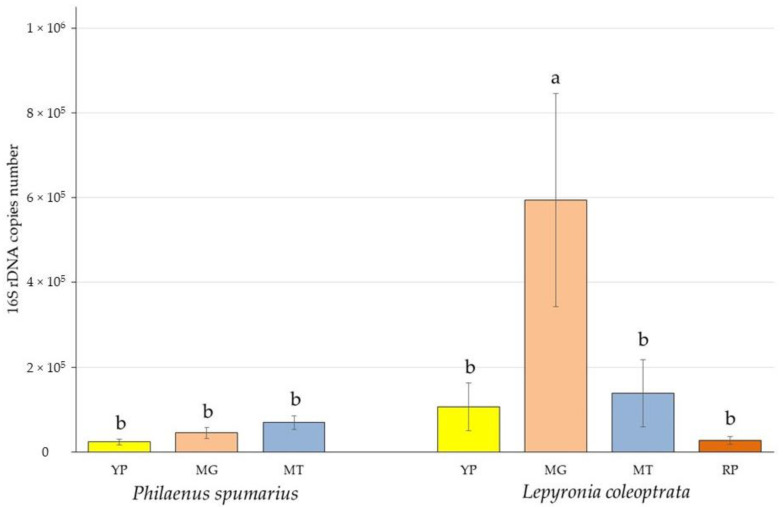
Mean number of 16S rDNA copies (bacterial biomass) measured for the PTM (posterior tubular midgut), ATM (filter chamber linked to the conical segment), MT (Malpighian tubules), and IL (ileum) of *P. spumarius* and *L. coleoptrata*. Bars indicate the standard error. Different letters above the histograms indicate significant differences among means (*p* < 0.05).

**Figure 2 microorganisms-11-00466-f002:**
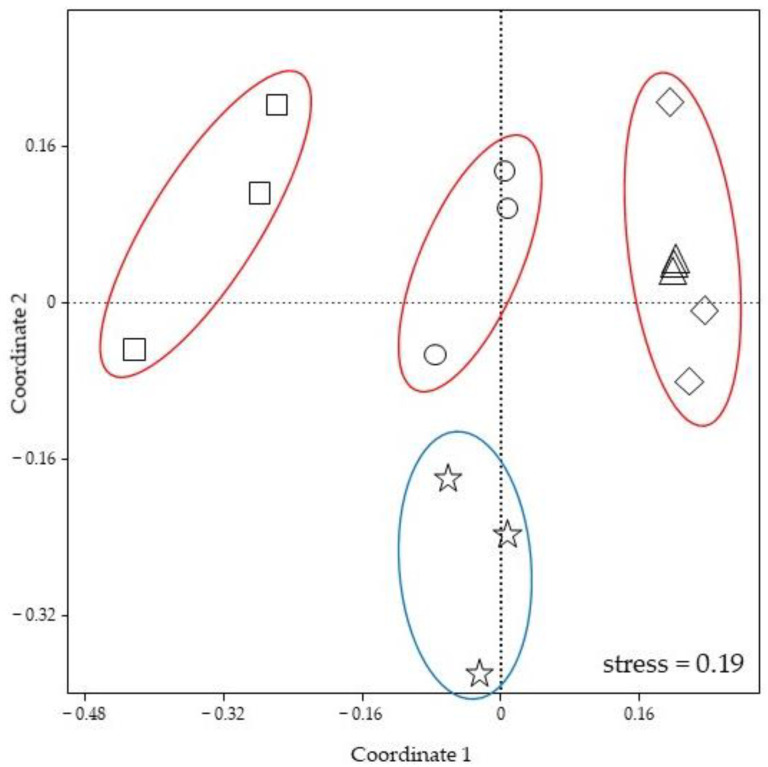
Non-metric multidimensional scaling (nMDS) ordination plot of bacterial communities detected in foams of *P. spumarius* and *L. coleoptrata*. Star: Foam of *L. coleoptrata* collected from *Trifolium repens*; Circle: Foam of *P. spumarius* collected from *Cirsium* sp.; Square: Foam of *P. spumarius* collected from *Cistus monspeliensis*; Diamond: Foam of *P. spumarius* collected from *Euphorbia cyparissias*; Triangle: Foam of *P. spumarius* collected from *Vicia sativa*.

**Table 1 microorganisms-11-00466-t001:** Sampling sites where nymphs of *Philaenus spumarius* and *Lepyronia coleoptrata* and/or their foams were collected from April to June 2021 (all sampling sites are located in Tuscany, Italy).

Site of Collection	GPS Coordinates	Species Nymph or Foam	Host Plant	Number of Collected Nymphs	Overall Amount of Collected Foam (mL)
Monte Argentario (Grosseto)	42.37701 N 11.18620 E	*P. spumarius*	*Cistus monspeliensis* (Cistaceae)	-	12
Fabio, Vaiano (Prato)	43.939682 N 11.142821 E	*P. spumarius*	*Vicia sativa* (Fabaceae)	-	9
			*Cirsium* sp. (Asteraceae)	-	9
Gavigno, Cantagallo (Prato)	44.040865 N 11.104997 E	*P. spumarius*	*Euphorbia cyparissias* (Euphorbiaceae)	-	10
			*Ranunculus sp.* (Ranunculaceae)	20	-
Montepaldi, San Casciano (Firenze)	43.667408 N 11.143868 E	*L. coleoptrata*	*Trifolium repens* (Fabaceae)	15	30

**Table 2 microorganisms-11-00466-t002:** Bacterial species detected in samples of foam produced by nymphs of *Philaenus spumarius* and *Lepyronia coleoptrata*, collected from different host plants. Identification of sequenced 16S rDNA bands selected from PCR-DGGEs.

Spittlebug Species	Host Plant	Bacterial Species	Class, Family	DGGE Band
*P. spumarius*	*Cistus monspeliensis*	*Ciceribacter selenitireducens*	α-Proteobacteria, Rhizobiaceae	F-11
	*Cistus monspeliensis*	*Ciceribacter selenitireducens*	α-Proteobacteria, Rhizobiaceae	F-13
	*Cistus monspeliensis*	*Ciceribacter azotofigens*	α-Proteobacteria, Rhizobiaceae	F-16
	*Cistus monspeliensis*	*Ciceribacter azotofigens*	α-Proteobacteria, Rhizobiaceae	F-17
	*Cistus monspeliensis*	*Brevundimonas mediterranea*	α-Proteobacteria, Caulobacteraceae	F-15
	*Cistus monspeliensis*	*Erwinia rhapontici*	γ-Proteobacteria, Enterobacteriaceae	F-14
	*Cistus monspeliensis*	*Stenotrophomonas rhizoplilia*	γ-Proteobacteria, Xanthomonadaceae	F-18
	*Cirsium* sp.	*Ciceribacter azotofigens*	α-Proteobacteria, Rhizobiaceae	F-31
	*Cirsium* sp.	*Pigmentiphaga humi*	β-Proteobacteria, Alcaligenaceae	F-10
	*Vicia sativa*	*Devosia oryziradicis*	α-Proteobacteria, Devosiaceae	F-12
*L. coleoptrata*	*Trifolium repens*	*Brevundimonas mediterranea*	α-Proteobacteria, Caulobacteraceae	F-26
	*Trifolium repens*	*Rhizobium skierniewicense*	α-Proteobacteria, Rhizobiaceae	F-28, F-29
	*Trifolium repens*	*Sinorhizobium* sp.	α-Proteobacteria, Rhizobiaceae	F-7
	*Trifolium repens*	*Erwinia rhapontici*	γ-Proteobacteria, Enterobacteriaceae	F-9

## Data Availability

Data that support the findings of this study are available on reasonable request from the corresponding author.

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
