# Peer review of "Diversity of the Bacterial Community Associated with Hindgut, Malpighian Tubules, and Foam of Nymphs of Two Spittlebug Species (Hemiptera: Aphrophoridae)"

_microorganisms, 2023, doi:10.3390/microorganisms11020466_

Round 1

Reviewer 1 Report

The paper is interesting, well written and organized, and the data look properly analyzed. Some notes are reported on the attached file

Author Response

Please, see the attached file

Reviewer 2 Report

In this study, the endosymbionts of P. spumarius nymphs were explored and nymphs of the spittlebug were examined to explore and compare the gut microbiota. The structure of the bacterial community associated with the foam and the Malpighian tubules involved in the froth production were studied. To provide some reference for the study of microbial polymorphism of foam-spitting spittlebug. But there are some unavoidable problems with the manuscript.

1.      Why did the authors choose fifth instar nymph in this study? It was not described in the manuscript.

2.      In the table describing sample collection in the method, I found that the foam and nymph collected by the author were collected from different places. How does the author consider this? We know that the gut microbes of insects are strongly influenced by the host plants and the environment. In this paper, foam and nymph in different environments were compared. There might be a big difference in microbial polymorphism. The reliability of the analysis in Figure 2 is questionable.

3.      The insects and foam in this experiment were collected directly from the field. How to distinguish the microbes that temporarily pass through the gut when the insects feed on the plants from the insect endosymbionts? In addition, the microorganisms in foam of Spittlebugs are more seriously affected by the environment. How to consider the influence of environmental microorganisms on the experimental results? For example, Rhodococcus and Escherichia coli mentioned in the article are very common in the environment.

4.      There was no description of the amount of foam repeated in the methods of this manuscript, only a reference to 40 mL of foam is found in line 208.

5.      In result 3.1, the richness in endosymbionts was analyzed and the variance analysis results showed a very large error. Is this result due to insufficient sample quantity? Would it be more appropriate to do the variance analysis by taking the logarithm method?

6.      The data of IL of P. spumarius in Figure 1 was not shown.

Author Response

Please, see the attached file

Round 2

Reviewer 2 Report

Accept in present form.